# Chemical Composition of the Essential Oil of *Catha edulis* Forsk from Djibouti and Its Toxicological Investigations In Vivo and In Vitro

Fatouma Mohamed Abdoul-Latif [1,*], Ayoub Ainane [2], Ibrahim Houmed Aboubaker [3], Ali Merito Ali [1], Zineb El Montassir [2], Mateusz Kciuk [4,5], Jalludin Mohamed [1] and Tarik Ainane [2]

[1] Medicinal Research Institute, Center for Studies and Research of Djibouti, IRM-CERD, Route de l'Aéroport, Haramous, Djibouti City P.O. Box 486, Djibouti
[2] Superior School of Technology of Khenifra, University of Sultan Moulay Slimane, P.O. Box 170, Khenifra 54000, Morocco
[3] Peltier Hospital of Djibouti, Djibouti City P.O. Box 2123, Djibouti
[4] Doctoral School of Exact and Natural Sciences, University of Lodz, Banacha Street 12/16, 90-237 Lodz, Poland
[5] Department of Molecular Biotechnology and Genetics, University of Lodz, Banacha 12/16, 90-237 Lodz, Poland
* Correspondence: fatoumaabdoulatif@gmail.com

**Abstract:** This work aimed to evaluate the chemical composition of essential oils from *Catha edulis* Forsk collected in the Day Forest of the Republic of Djibouti. Additionally, in vivo toxicity studies, biochemical profiling, and hematological tests were conducted to determine the biological activity of the investigated essential oils. Finally, in vitro tests were performed to investigate the antibacterial activity of the essential oils. The essential oils were obtained at yields of 0.75%. Chromatographic analysis identified 39 compounds, of which cathinone (81.4%) and cathine (10.55%) were determined as the two major components, representing 91.95% of the total composition. *Catha edulis* essential oil had a rat LD50 of 2500 mg/kg, indicating very low toxicity. Chronic exposure studies revealed that use of the essential oil in rats resulted in persistent stimulant action at dosages of 100 and 200 mg/kg, whereas the weight gain of control rats was faster than that of the essential oil-treated rats. Hematological parameters showed a significant increase in red blood cell counts, hemoglobin, hematocrit, and platelets in treated rats, which could indicate blood hyperviscosity that may increase the risk of blood clots and cardiovascular diseases. Furthermore, the investigated essential oil exhibited antimicrobial activity against several bacterial strains.

**Keywords:** essential oil; *Catha edulis* Forsk; chemical composition; biological activities; toxicity; hematology; biochemical tests; antibacterial activity

## 1. Introduction

The khat plant (*Catha edulis* Forsk) is a tree from the Celastraceae family, commonly cultivated in parts of East Africa and the Arabian Peninsula [1]. The plant's leaves contain alkaloids that are structurally similar to amphetamines and are chewed daily by a high proportion of Djibouti's adult population due to their mild stimulating effect. Initially, khat was used as a drink made from dried leaves, but it was later discovered that chewing the green leaves produced a stronger effect due to the loss of active constituents during the drying process [2,3]. Chewing khat leaves has been a common practice in Djibouti for hundreds of years. It is a major stimulant that helps people work harder, fights fatigue, and helps students study for tests. Recently, the consumption of fresh khat leaves has expanded due to air transport improvements, even in communities in Europe [4]. Chemical analysis has confirmed that the fresh leaves contain phenylalkylamine compounds (alkaloids) such as cathine and beta-cathinone, which are similar to amphetamines. Khat leaves also contain tannins, vitamins, minerals, and flavonoids. Currently, cathinone is believed to be the main

active ingredient in fresh khat leaves [5–7]. Proponents of chewing khat claim that it has various health benefits, such as lowering blood sugar levels, acting as a remedy for asthma, and relieving symptoms of bowel tract disorders [8]. However, opponents argue that khat can harm health and affect many aspects of life, leading to social, economic, and medical consequences. In Djibouti, due to the widespread use of khat, it has become a serious national concern [9,10].

The objective of this study was to evaluate the chemical composition of essential oils extracted from *Catha edulis* Forsk collected in the Day Forest of the Republic of Djibouti. Additionally, the study aimed to determine the biological activity of the essential oils through in vivo toxicity studies, biochemical profiling, and hematological tests, as well as to investigate their antimicrobial activity through in vitro tests.

## 2. Materials and Methods

### 2.1. Plant Material and Sample Collection

The *Catha edulis* Forsk plant, which has medicinal properties, belongs to the Celastraceae family. In March 2022, the aerial parts of the plant were collected from the Day Forest in the Republic of Djibouti and then dried in the shade. The authentication of the plant species was performed by the first and last author of the work in the herbarium of the Institut de Recherches Médicinales de Djibouti (CERD) with the accession number CE11-2022.

### 2.2. Botanical Description of Catha edulis Forsk

Khat is an evergreen tree that is glabrous and upright and can grow to a height of 25 m. Its branches are dimorphic, with a small pointed crown at the end [11]. However, when cultivated, it is a shrub with multiple stems and a maximum height of 6 m. The trunk is straight and slender, and its bark differs between its natural and cultivated states. In cultivation, the bark is smooth and pale green–gray, while in large trees, it is rough. The branches are cylindrical and grayish to brownish [12]. Young twigs are usually flattened and have a dull green or brownish-red color. The leaves of the khat plant are alternate on orthotropic branches and opposite on plagiotropic branches. The triangular stipules are pale green, measure 3 cm by 1 mm, and are deciduous, leaving a beaded scar. The cylindrical petiole is 3 to 11 mm long and varies in color from pale to dark green. The leaf blade is oblong, elliptical, or obovate, with a size range of $5.5 \times 1.5$ cm to $11 \times 6$ cm. The base of the leaf is wedge-shaped to attenuate, and the apex is acute to acuminate or even obtuse. The edges of the leaf are glandular, serrated, and shiny. The mature leaf is leathery with reticulate venation [13].

### 2.3. Extraction and Analysis of Essential Oil

A Clevenger-type apparatus was used in a 4 h hydrodistillation process to obtain the essential oil from 500 g of dried *Catha edulis* Forsk plant parts. The resulting essential oil was collected and dried using anhydrous sodium sulfate, after which it was stored at 4 °C until analysis. The oil yield was determined based on the plant's dry weight [14,15].

The yield of essential oil, represented as (Y), is determined by dividing the volume of extracted oil (v) by the weight of plant material used (W). This value is typically expressed in mL distillate per 100 g of dry matter and represented as a percentage of volume to mass, denoted as $v/w\%$ [16,17].

$$Y(\%) = \frac{v}{W} \times 100$$

Gas chromatography coupled with mass spectrometry (GC-MS) is a commonly used analytical technique for the identification and quantification of volatile compounds in essential oils. In this analysis of *Catha edulis* Forsk essential oil, an Agilent GC-MS system was used. An Agilent Gas Chromatograph (GC-FID) model 6890 was utilized in the system and was equipped with a 30 m-long HP-5 MS silica capillary column. The column had an internal diameter of 0.25 mm and a film thickness of 0.25 μm. The stationary phase was

composed of 5%-phenyl methylpolysiloxane. The column was programmed to increase from 50 °C (held for 5 min) to 250 °C at a rate of 5 °C/min and then held for 5 min. The injector temperature and flame ionization detector temperature were set to 280 °C and 300 °C, respectively. For analysis, 1 µL of essential oil diluted in 3.5% (*v/v*) acetone was injected in split mode (1/60). Hydrogen was used as the carrier gas with a flow rate of 1.0 mL/min [18–20].

### 2.4. Toxicity Study

#### 2.4.1. Experimental Animals

The tests were performed on Swiss mice weighing 25–35 g and adult male Wistar rats weighing between 180–220 g. However, the tests to calculate the $LD_{50}$ were carried out on female Swiss mice weighing 20–30 g. The animals were housed in the animal experimentation center of the faculty and kept under standard environmental conditions, including regulated temperature, relative humidity, and light exposure (26 ± 1 °C, 60–70% humidity and a 12 h light–12 h dark cycle) [21]. Food and water were provided ad libitum, except during the fasting period before the experimental procedures. The treatment of the animals strictly adhered to the guidelines stated in the *Official Journal of the European Community* for the care and use of laboratory animals [22]. Ethical approval for this study was granted by the ethics committee of ESTK-USMS (Morocco) and biological data were archived under accession number 2022/1000.

#### 2.4.2. Study of the Acute Toxicity of the Essential Oil of *Catha edulis* Forsk

We used the acute oral toxicity method from OECD guideline 423 (OECD, 2002) to test the short-term toxicity of *Catha edulis* Forsk essential oil. This method is reproducible, uses very few animals, and is different from other acute toxicity methods (Guidelines 420 and 425). It allows substances to be classified by toxicity order. A determined dose of the substance was administered orally to a group of animals.

The experiment was carried out for each step on three non-pregnant female Swiss mice aged between 2 and 2.5 months and weighing between 20 and 30 g, which were kept healthy and fasted (food was removed but not water) for 4 h before and 2 h after oral administration of *Catha edulis* Forsk essential oil with an initial dose of 300 mg/kg. The essential oil was administered in solution in peanut oil by the oral route at a volume of 0.5 mL per 20 g of body weight. The treated mice were kept under continuous observation on the day of gavage to note any immediate disturbance. Observation of weight variations, mortality rates, animal behavior, and signs of toxicity was also carried out during the 14 days following exposure to detect any delayed effects of the extracts [23–25].

#### 2.4.3. Study of the Chronic Toxicity of the Essential Oil of *Catha edulis* Forsk

We conducted a 90-day chronic toxicity study via oral administration following OECD Guidelines 408 and 452. The Wistar strain rats used in the study were bred in the central animal facility of the faculty. They were divided into six groups of twelve rats each, with each group composed of six males and six females housed separately in cages (females separated from males). The rats weighed between 180 and 200 g, pre-marked, and kept under standard conditions (temperature of 26 ± 1 °C, humidity of 60–70%, and a 12 h light–12 h dark cycle). All animals had free access to water and were acclimated for five days before the start of the study. One group of control rats received peanut oil instead of *Catha edulis* Forsk essential oil extract, while the other groups received the tested *Catha edulis* Forsk oil extract. The oil extract used in our study contained diluted *Catha edulis* Forsk essential oil in peanut oil at doses of 100 and 200 mg/kg administered orally. These doses were below the $LD_{50}$ and represented a tenth of this value, corresponding to 100 and 200 mg/kg for *Catha edulis* Forsk essential oil, respectively. Throughout the exposure period, the animals' body weight was monitored by weighing them weekly and noting their behavior. We also took blood samples at four different times (T0, T1, T2, and T3) to perform hematological and biochemical tests to evaluate the extracts' impact on certain organs and animal metabolism.

The samples were taken after a brief ether anesthesia on an empty stomach, through the retro-orbital sinus at eye level using a heparinized Pasteur pipette. Blood was collected in EDTA tubes (for hematological parameters) and heparinized tubes (for biochemical analysis) for the analysis of hematological and biochemical parameters [26–28].

### 2.4.4. Hematological Tests

The blood samples collected on EDTA were immediately used to measure levels of white blood cells, red blood cells, platelets, leukocyte formula, hemoglobin, and hematocrit. At the end of the experiment, all surviving animals were euthanized under ether anesthesia. The heart, lungs, liver, kidneys, and spleen were removed for histopathological analysis. This analysis aimed to identify any lesions caused by the treatment on these organs through macroscopic and microscopic analyses. After removal, each organ was weighed and preserved in a flask containing 10% formalin to prevent drying. Each organ underwent microscopic analysis. Hematological and biochemical measurements, as well as histopathological analyses, were performed in the laboratory [29,30].

### 2.4.5. Biochemical Tests

The blood samples collected in heparinized tubes were centrifuged at 4000 revolutions per minute for 10 min. The obtained serum was used to measure various biochemical parameters, including glucose, creatinine, urea, alanine aminotransferase (ALT), and aspartate aminotransferase (AST) levels [31,32].

### 2.5. *In Vivo Activities of Catha edulis Forsk Essential Oil*

The determination of the psychopharmacological profile of a substance requires an array of pharmacological tests to evaluate its impact on the nervous system. The study methods are classified into three groups: behavioral, neurochemical, and electrophysiological. These three methods are complementary in defining the psychopharmacological profile of a substance. In this study, behavioral methods were used to evaluate the psychotropic activity of an oily extract. The results of these tests helped to determine the potential clinical use of *Catha edulis* Forsk essential oil. In this experimental design, bromazepam (20 mg/kg, p.o.) was used as a reference drug [33–35].

### 2.5.1. Traction Test

During a test, mice were hung by their front legs from a wire that was stretched out horizontally. Typically, normal mice will quickly readjust themselves and reach the wire within 5 s. If a mouse fails to reach the wire using one of its hind legs, it is considered to be under the influence of a sedative, and this is classified as a negative reaction. On the other hand, if the mouse can reach the wire with one of its hind legs, it is classified as a positive reaction. The behavior of the mice was observed and recorded throughout the test [36].

### 2.5.2. Chimney Test

The experiment involved placing a rat in a 30 cm-long vertical glass tube and observing its attempts to climb backward. The maximum time a mouse was allowed to escape was 120 s, and a normal mouse usually escaped within 30 s. Mice that took longer than 30 s to respond were considered sedative-affected [37].

### 2.5.3. Hole-Board Test

This test allows the study of the exploration behavior of mice in a new environment and is used to highlight the inhibitory effect of psycholeptics on this reaction. The hole-board model LE-8825, measuring 40 cm × 40 cm by 2.2 cm thick and comprised of 16 holes of 3 cm diameter, was used in the study. Hole exploration movements were detected by infrared cells, and the counting of the number of holes explored was displayed every minute for 5 min. The mice were kept in their cages for up to 30 min after the administration of the essential oil (100, 200, 400, and 500 mg/kg; VO) to avoid any prior curiosity. The

observer remained silent and still throughout the trial. This method was described by Boissier et al. in 1964 [38].

### 2.5.4. Rota-Rod Test

The rotarod test was performed using an LE 8500 Rota Rod/RS apparatus consisting of a 50 cm-long and 3 cm-diameter horizontal Perspex pole that provided optimal grip for rodents. Vertical partitions provide four isolated compartments that allow the simultaneous assessment of four animal subjects. The pole was located 25 cm above pallets mounted on an axis, allowing for their inclination and the stopping of the timer when the animal fell. The pole rotates freely around its longitudinal axis thanks to a DC motor. The rotation speed of the axis can be modified in a range from 4 to 40 rpm, and it is possible to use it with a constant rotation speed. The day before the rotarod test, mice were subjected to a pretest that involved placing them on the pole and exposing them to rotation at a speed of 12 revolutions per minute to select those that would remain on the bar for at least 60 s. On the following day, extracts were injected into preselected mice 30 min before the test. Motor incapacity was evaluated at 30, 60, and 120 min after oral administration of the essential oil (100, 200, 400, and 500 mg/kg; VO). The time taken by the mice before falling from the rotarod was recorded. For each test, three batches of five mice were used: a control group receiving the solvent of the extract, a reference group receiving bromazepam (20 mg/kg), and a group treated with the extracts [39].

### 2.6. Evaluation of the Antibacterial Activity

The antimicrobial efficacy of *Catha edulis* Forsk essential oils was evaluated against six susceptible microbial strains. The strains used included four Gram-negative bacteria—*Escherichia coli*, *Pseudomonas aeruginosa*, *Proteus mirabilis,* and *Enterobacter cloacae*—as well as two Gram-positive bacteria: *Streptococcus pneumoniae* and *Staphylococcus aureus*.

### Antimicrobial Tests

We used the disk diffusion method to evaluate the antibacterial activity of plant extracts. The aromatogram method previously described by Jacob and Tonei in 1979 was employed in the study [40]. This method involves using sterile paper disks of 6 mm impregnated with different concentrations of extract obtained by a series of dilutions. Sterile Müller–Hinton agar was used to pour sterile petri dishes of 9 cm, on which an inoculum prepared from each strain was evenly spread. Paper disks impregnated with the extract were positioned on the agar surface and incubated at 37 °C for 18 h. A clear circular zone around the disk indicated the absence of bacterial growth. The diameter of this zone was measured with a ruler to determine the antimicrobial activity, which was expressed in terms of the diameter of the zone of inhibition (DZI). Negative controls (DMSO and distilled water) and positive control (ciprofloxacin) were also used to validate the results. The degree of sensitivity measured by the DZI was recorded as follows: extremely sensitive (+++) for D > 20 mm, very sensitive (++) for D between 15 and 19 mm, sensitive (+) for D between 9 and 14 mm, and nonsensitive (-) for D < 8 mm.

### 2.7. Statistical Analysis

Statistical analysis was performed using the XLSTAT toolbox (2016), which is part of the Microsoft Excel software. Values are presented as the mean ± uncertainty at a 5% significance level for each experiment, with three replicates conducted, and the Student's *t*-test used for statistical analysis. Analysis of variance (ANOVA) with Tukey's test was used to initially determine any significant data differences between the groups of samples.

## 3. Results and Discussion

### 3.1. Extraction and Analysis of Essential Oil

The average essential oil yield of *Catha edulis*, which was calculated based on the dry weight of plant material obtained from the aerial parts (stems, leaves, and flowers) of the studied plant, was estimated to be 0.75%.

However, this rate is lower than that obtained by hydrodistillation of the dried aerial parts of *Catha edulis* Forsk from Yemen, which is 0.9%, as described by Algabr et al. [41]. On the other hand, the essential oil yields obtained by hydrodistillation of the leaves of the two Ethiopian khat varieties are very low, less than 0.01% of the dry weight, as indicated by Hailu et al. [42]. It is therefore possible to conclude that several factors could be responsible for these variations, including the age of the plants, the age of the tree, the nature of the soil and climate, as well as the part of the plant used for extraction and the harvest period.

The results of gas chromatography coupled with mass spectrometry analysis of *Catha edulis* Forsk essential oil are presented in Table 1. Chromatographic analyses of the essential oil allowed for the identification of 39 compounds, which represent approximately 99.95% of the total composition. Cathinone (81.4%) and cathine (10.55%) were identified as the two major compounds. The remaining constituent content is low. It is important to note that cathinone is a stimulant alkaloid that is considered the most important psychoactive substance of the *Catha edulis* Forsk plant, while cathine is also a stimulant alkaloid, but is considered less psychoactive than cathinone. In general, the effects of consuming *Catha edulis* Forsk are due to the presence of these two compounds. These results may have implications for the consumption of *Catha edulis*, as the high concentration of cathinone may be associated with more intense psychostimulant effects. Additionally, the low content of other constituents may indicate that the plant does not have significant medicinal properties beyond its stimulant effects [43–45].

According to Hailu et al. (2017) [42], the compositions of the essential oils of khat from two different parts of Ethiopia were studied. The results showed that 50 compounds were identified in the essential oil of khat from Bahir Dar, and the major constituents were 1-phenyl-1,2-propanedione (11.6%), limonene (30%), O-mentha-1(7),8-dien-3-ol (8.5%), camphor (6.3%), Z-caryophyllene (3.9%), 3-carene (2.2%), β-bourbonene (2.4%), β-humulene (1.6%), and α-cubebene (1.5%). On the other hand, 34 compounds were identified in the essential oils of khat from Wendo, and the major components were hexyl pentyl ether (2.2%), 1-hydroxy,1-phenyl-2-propanone (8.1%), tritriacontane (12%), 3,9-dimethyl-undecane (2.5%), 2,2,8-trimethyl-decane (2.4%), hexadecane (5.7%), sulfurous acid ester-2-propylundecyl (5.5%), 2-ethyl-1-dodecanol (2.6%), heptadecane (5.1%), 2,10-dimethyl-9-undecenol (4%), 10-methylnonadecane (5.3%), nonadecane (4.4%), butyltetradecyl phthalate (2.8%), isobutyloctadecyl phthalate (6%), and (E)-5-eicosene (2.6%). Algabr et al. (2014) [41] reported that the compositions of the essential oils obtained from khat grown in the two regions of Ethiopia differ from each other as well as from those of Yemen. The study revealed the presence of 56 constituents, accounting for 99.9% of the total content, out of which only 37 compounds were identified. Carvotanacetone was the major constituent, constituting 84.41% of the essential oil.

### 3.2. Acute Toxicity of Catha edulis

The $LD_{50}$ of *Catha edulis* Forsk essential oil was determined using the method described in the European OECD Guideline 423. At a dose of 300 mg/kg administered orally, the mice showed slight sedation without any other notable effects. However, at a dose of 2000 mg/kg, only one mouse died shortly after administration, exhibiting signs of discomfort, such as accelerated breathing, agitation, and tremors. The surviving mice did not feed, but had recovered their normal state by the second day. The weight of the mice was not significantly affected by the extracts during the 14 days following treatment. Additionally, stimulant effects were observed in treated mice. As a result, the $LD_{50}$ for *Catha edulis Forsk* essential oil was estimated to be 2500 mg/kg and was categorized as a toxic compound of class V.

**Table 1.** Chemical composition of *Catha edulis* Forsk essential oil.

| Peak | Retention Time | Compound | Percentage (%) |
|---|---|---|---|
| 1 | 9.88 | Dehydrocineole | 0.03 |
| 2 | 10.04 | A-Phellandrene | 0.03 |
| 3 | 10.23 | Dolcymene | 0.10 |
| 4 | 10.29 | B-Phellandrene | 0.07 |
| 5 | 10.31 | Eucalyptol | 0.02 |
| 6 | 10.97 | Linalol | 0.13 |
| 7 | 11.23 | Cis-2-Menthenol | 0.08 |
| 8 | 11.42 | Menth-2-En-1-Ol Trans | 0.06 |
| 9 | 11.47 | Camphre | 0.19 |
| 10 | 11.70 | Isoborneol | 0.07 |
| 11 | 11.72 | Endo Borneol | 0.35 |
| 12 | 11.96 | A-Terpineol | 0.12 |
| 13 | 12.02 | Tetrahydrocarvone | 0.29 |
| 14 | 12.10 | Trans Pulegol | 0.16 |
| 15 | 12.27 | Thymol Methyl Ether | 0.56 |
| 16 | 12.37 | Cis Carvotanacetol | 0.78 |
| 17 | **12.56** | **Cathinone** | 81.40 |
| 18 | 12.59 | Carvenone | 0.92 |
| 19 | 12.85 | Acetate De Bornyle | 0.04 |
| 20 | 12.90 | Thymol | 0.08 |
| 21 | 12.96 | Carvacrol | 0.48 |
| 22 | 13.50 | Eugenol | 0.32 |
| 23 | 13.78 | Damascenone | 0.07 |
| 24 | 14.01 | Thymoquinol Dimethylether | 0.59 |
| 25 | 14.21 | B-Caryophyllene | 0.05 |
| 26 | 14.45 | Geranyl Acetone | 0.04 |
| 27 | 14.74 | Minacide | 0.75 |
| 28 | 14.77 | (E)-B-Ionone | 0.10 |
| 29 | 14.82 | Neryl Isobutyrate | 0.48 |
| 30 | 15.18 | Δ-Cadinene | 0.08 |
| 31 | 15.59 | (E)-Nerolidol | 0.04 |
| 32 | 15.68 | Geranyl Isobutyrate | 0.22 |
| 33 | 15.76 | Neryl 2-Methylbutyrate | 0.07 |
| 34 | 15.84 | Oxide De Caryophyllene | 0.30 |
| 35 | 16.42 | Epi-A-Cadinol | 0.04 |
| 36 | 16.44 | Epi-A-Muurolol | 0.03 |
| 37 | 16.55 | A-Cadinol | 0.14 |
| 38 | 19.53 | Phthalate | 0.12 |
| 39 | **19.67** | **Cathine** | 10.55 |
| | **Total** | | **99.95%** |

### 3.3. Chronic Toxicity of Catha edulis Forsk

The essential oil of *Catha edulis* Forsk is considered to have very low toxicity. For this reason, we evaluated its chronic toxicity using experimental therapeutic doses to assess the impact of this essential oil after repeated administration. The results of this evaluation were divided into five points, as described below.

#### 3.3.1. Animal Behavior

Rats treated with essential oil at a dose of 100 mg/kg exhibited normal behavior. On the other hand, those treated with a dose of 200 mg/kg showed prolonged stimulant activity for several hours, followed by a period of slowed movement and sedation combined with isolation, which lasted about 15 min, before returning to their normal state.

#### 3.3.2. Weight Change

The results of the weight evaluation of two groups of treated animals are presented in Table 2. Compared to the body weight on the first day of the experiment, a significant and normal increase in weight was observed after 90 days in both control rats and those treated with respective doses of 100 and 200 mg/kg. However, a difference was noted between the two groups, with animals treated with doses of 100 and 200 mg/kg having lower weights than untreated animals. These weight changes are statistically significant ($p < 0.05$). It was observed that the weight gain of control rats was faster than that of treated rats, confirming the inhibitory effect of *Catha edulis* Forsk essential oil on weight gain.

**Table 2.** Body weight of rats treated with *Catha edulis* Forsk essential oil.

| Weeks | Control | 100 mg/kg | 200 mg/kg | ANOVA | |
|---|---|---|---|---|---|
| | | | | F-Ratio | *p*-Value |
| W0 | 184.54 ± 10.11 | 184.27 ± 10.94 | 184.62 ± 12.06 | 0.02 | 0.71 |
| W1 | 200.21 ± 9.55 | 195.38 ± 9.67 | 195.02 ± 12.58 | 0.59 | 0.50 |
| W2 | 217.49 ± 8.93 | 206.64 ± 8.45 | 206.37 ± 12.36 | 3.12 | 0.11 |
| W3 | 226.55 ± 9.68 | 220.73 ± 9.37 | 220.11 ± 12.64 | 0.36 | 0.57 |
| W4 | 231.39 ± 10.74 | 227.41 ± 10.69 | 226.19 ± 13.25 | 1.14 | 0.36 |
| W5 | 235.91 ± 12.11 | 229.79 ± 12.67 | 228.38 ± 13.80 | 1.25 | 0.33 |
| S6 | 239.75 ± 12.96 | 234.43 ± 12.73 | 232.97 ± 13.52 | 0.24 | 0.62 |
| W7 | 248.30 ± 12.64 | 239.11 ± 12.36 | 238.76 ± 13.34 | 1.08 | 0.37 |
| W8 | 255.63 ± 12.45 | 249.71 ± 12.49 | 248.67 ± 13.67 | 0.50 | 0.51 |
| W9 | 261.11 ± 12.95 | 257.11 ± 12.39 | 256.19 ± 13.74 | 0.41 | 0.56 |
| W10 | 268.25 ± 13.45 | 265.37 ± 13.97 | 262.74 ± 13.67 | 0.34 | 0.58 |
| W11 | 272.81 ± 13.15 | 268.46 ± 13.89 | 266.61 ± 13.79 | 0.63 | 0.49 |
| W12 | 278.44 ± 13.67 | 274.85 ± 13.81 | 269.82 ± 13.26 | 0.74 | 0.45 |
| Weight gain | 93.90 ± 0.5 [a] | 90.58 ± 0.45 [b] | 85.2 ± 0.48 [c] | 10.21 | <0.05 * |

Different letters in the same row indicate significant differences according to Tukey's test ($p < 0.05$). * Values are significant at $p < 0.05$.

#### 3.3.3. Hematological Parameters

The results of the hematological parameters are presented in Table 3. The levels of red blood cells (RBC), hemoglobin (HGB), hematocrit (HCT), and platelets (PLT) increased significantly, which may be a sign of blood hyperviscosity and increased risk of blood clots and cardiovascular diseases [46]. However, these parameters should be interpreted in conjunction with other risk factors to assess the individual risk of cardiovascular disease [47]. On the other hand, no significant differences were observed for other parameters in rats treated compared to control groups, even as the duration of treatment increased. The



hematological parameters remained the same in the two groups of treated rats compared to the results of the control group.

**Table 3.** Effects of repeated oral administration of *Catha edulis* Forsk essential oil for 90 days on hematological parameters in Wistar rats.

| Settings | Control | Processing Time | | | | | | ANOVA | |
| | | 30 Days | | 60 Days | | 90 Days | | F-Ratio | *p*-Value |
| | | 100 (mg/kg) | 200 (mg/kg) | 100 (mg/kg) | 200 (mg/kg) | 100 (mg/kg) | 200 (mg/kg) | | |
| --- | --- | --- | --- | --- | --- | --- | --- | --- | --- |
| HGB (g/dL) | 14.21 ± 0.46 | 14.38 ± 0.94 | 14.56 ± 0.95 | 14.59 ± 0.95 | 14.85 ± 0.93 | 14.77 ± 0.94 | 15.21 ± 0.95 | 0.22 | 0.62 |
| GR ($10^6/\mu$L) | 7.45 ± 0.64 | 7.52 ± 0.34 | 7.58 ± 0.72 | 7.83 ± 0.36 | 8.21 ± 0.43 | 8.08 ± 0.70 | 8.84 ± 0.53 | 3.02 | 0.11 |
| HCT (%) | 40.11 ± 0.52 | 41.16 ± 3.44 | 42.31 ± 2.83 | 41.35 ± 3.71 | 42.41 ± 3.99 | 42.70 ± 3.35 | 42.49 ± 3.41 | 3.21 | 0.10 |
| PLQ ($10^3/\mu$L) | 625 ± 57 | 634 ± 59 | 648 ± 61 | 641 ± 60 | 652 ± 59 | 645 ± 57 | 657 ± 60 | 0.17 | 0.64 |
| GB ($10^3/\mu$L) | 10.63 ± 2.37 | 10.59 ± 2.29 | 10.47 ± 2.31 | 10.56 ± 2.46 | 10.55 ± 2.17 | 10.49 ± 2.29 | 10.49 ± 2.64 | 0.12 | 0.67 |
| NEUT (%) | 20.11 ± 0.05 [a] | 20.11 ± 0.09 [a] | 20.01 ± 0.09 [a] | 21.03 ± 0.1 [b] | 20.12 ± 0.08 [a] | 20.15 ± 0.1 [a] | 20.09 ± 0.1 [a] | 6.11 | <0.05 * |
| LYMPH (%) | 69.99 ± 1.34 | 69.72 ± 2.72 | 69.55 ± 3.22 | 69.98 ± 1.91 | 69.21 ± 3.17 | 69.54 ± 2.43 | 69.75 ± 3.21 | 0.15 | 0.66 |
| MONO (%) | 2.03 ± 0.99 | 1.97 ± 0.98 | 1.97 ± 0.99 | 1.83 ± 1.03 | 1.99 ± 1.00 | 1.82 ± 1.04 | 2.02 ± 1.02 | 0.94 | 0.40 |
| EO (%) | 1.69 ± 0.41 | 1.69 ± 0.53 | 1.70 ± 0.50 | 1.71 ± 0.73 | 1.69 ± 0.88 | 1.68 ± 1.16 | 1.70 ± 1.31 | 0.77 | 0.45 |
| MCV (fL) | 51.22 ± 3.41 | 52.72 ± 2.37 | 52.75 ± 1.93 | 52.15 ± 2.18 | 50.23 ± 2.34 | 52.17 ± 2.64 | 52.16 ± 3.43 | 0.71 | 0.46 |
| MCHC (g/dL) | 36.09 ± 0.94 | 36.15 ± 1.00 | 36.25 ± 1.01 | 36.21 ± 0.59 | 36.17 ± 1.11 | 36.19 ± 1.05 | 36.18 ± 1.00 | 0.07 | 0.68 |
| MCH (pg) | 18.42 ± 0.94 | 17.68 ± 0.61 | 18.47 ± 0.79 | 17.68 ± 0.97 | 18.43 ± 0.85 | 18.00 ± 0.67 | 18.32 ± 1.00 | 0.34 | 0.59 |

Different letters in the same row indicate significant differences according to Tukey's test ($p < 0.05$). * Values are significant at $p < 0.05$.

### 3.3.4. Biochemical Parameters

After analyzing the results presented in Table 4, it was observed that the levels of ALAT, ASAT, triglycerides, and total proteins did not undergo significant changes in either treated group compared to the controls. However, the levels of urea and creatinine significantly increased in both groups of animals, while the cholesterol level significantly decreased. In addition, the blood glucose level remained stable. These results suggest that *Catha edulis* Forsk essential oil may reduce the level of blood cholesterol (HDL), thereby increasing the risk of cardiovascular disease [48]. However, an increase in the levels of urea and creatinine may indicate kidney dysfunction or dehydration. Although the khat plant has been associated with stabilizing blood sugar levels, surveys have revealed that consumers of this plant suffer from hyperglycemia due to the consumption of sugary drinks with this plant to improve taste, as the plant's leaves cause dryness in the mouth [49,50].

**Table 4.** Effects of repeated oral administration of *Catha edulis* Forsk essential oil for 90 days on biochemical parameters in Wistar rats.

| Settings | Control | Processing Time | | | | | | ANOVA | |
| | | 30 Days | | 60 Days | | 90 Days | | F-Ratio | *p*-Value |
| | | 100 (mg/kg) | 200 (mg/kg) | 100 (mg/kg) | 200 (mg/kg) | 100 (mg/kg) | 200 (mg/kg) | | |
| --- | --- | --- | --- | --- | --- | --- | --- | --- | --- |
| ALAT (U/I) | 67.89 ± 4.81 | 67.75 ± 2.31 | 66.85 ± 4.75 | 67.57 ± 1.75 | 66.99 ± 3.34 | 67.87 ± 2.97 | 67.68 ± 3.55 | 0.24 | 0.62 |
| ASAT (U/I) | 223.66 ± 15.24 | 220 ± 25.17 | 221 ± 20.67 | 222 ± 26.37 | 223 ± 25.87 | 221 ± 25.37 | 222 ± 23.37 | 0.37 | 0.56 |
| Cholesterol (g/L) | 0.86 ± 0.51 | 0.86 ± 0.53 | 0.86 ± 0.56 | 0.81 ± 0.37 | 0.79 ± 0.39 | 0.80 ± 0.25 | 0.77 ± 0.24 | 0.33 | 0.58 |
| Triglycerides (g/L) | 0.66 ± 0.52 | 0.66 ± 0.54 | 0.66 ± 0.67 | 0.66 ± 0.57 | 0.66 ± 0.49 | 0.66 ± 0.47 | 0.66 ± 0.61 | 0.04 | 0.71 |
| Creatinine (mg/L) | 4.91 ± 0.52 [a] | 5.1 ± 0.59 [a] | 5.5 ± 0.81 [a] | 6.28 ± 0.69 [b] | 6.32 ± 0.82 [b] | 6.71 ± 0.53 [b] | 6.83 ± 0.73 [b] | 5.22 | <0.05 * |
| Urea (mg/L) | 0.25 ± 0.05 [a] | 0.32 ± 0.05 [b] | 0.36 ± 0.05 [b] | 0.36 ± 0.05 [b] | 0.41 ± 0.05 [b] | 0.40 ± 0.05 [b] | 0.45 ± 0.05 [b] | 5.47 | <0.05 * |
| Protein T (g/dL) | 67.65 ± 4.26 | 67.83 ± 3.65 | 67.16 ± 3.06 | 66.35 ± 6.62 | 67.67 ± 3.34 | 67.73 ± 1.51 | 67.83 ± 1.83 | 0.33 | 0.57 |
| Glucose (g/L) | 1.33 ± 0.12 [a] | 1.19 ± 0.09 [a] | 1.19 ± 0.05 [a] | 1.10 ± 0.02 [b] | 1.10 ± 0.01 [b] | 1.10 ± 0.01 [b] | 1.10 ± 0.01 [b] | 5.03 | <0.05 * |

Different letters in the same row indicate significant differences according to Tukey's test ($p < 0.05$). * Values are significant at $p < 0.05$.

### 3.3.5. Anatomopathological Exam

After macroscopic observation of the organs (lungs, heart, liver, kidneys, and spleen), no morphological anomalies or hemorrhage due to the administration of *Catha edulis* Forsk essential oil were revealed. Statistical analysis of organ weights in groups treated with doses of 100 and 200 mg/kg showed no significant difference compared to the control group (Table 5). Histological examination of tissues from the lungs, heart, liver, and spleen revealed no characteristic pathology or abnormalities in cell architecture compared to controls. However, at the level of the kidneys, we observed glomerular ischemia and focal tubular necrosis (Figure 1).

**Table 5.** Effect of *Catha edulis* Forsk essential oil on the weight of organs removed from rats after 90 days of oral treatment.

| Organs | Control | 100 (mg/kg) | 200 (mg/kg) | ANOVA | |
| --- | --- | --- | --- | --- | --- |
| | | | | F-Ratio | *p*-Value |
| Heart | $0.85 \pm 0.06$ | $0.85 \pm 0.07$ | $0.86 \pm 0.07$ | 0.24 | 0.60 |
| Liver | $10.52 \pm 0.24$ | $10.50 \pm 0.07$ | $10.53 \pm 0.12$ | 0.28 | 0.59 |
| Spleen | $0.72 \pm 0.05$ | $0.73 \pm 0.05$ | $0.73 \pm 0.05$ | 0.01 | 0.79 |
| Lungs | $1.79 \pm 0.05$ | $1.80 \pm 0.05$ | $1.79 \pm 0.05$ | 0.14 | 0.66 |
| Kidneys | $0.82 \pm 0.01$ | $0.83 \pm 0.01$ | $0.82 \pm 0.01$ | 0.03 | 0.71 |

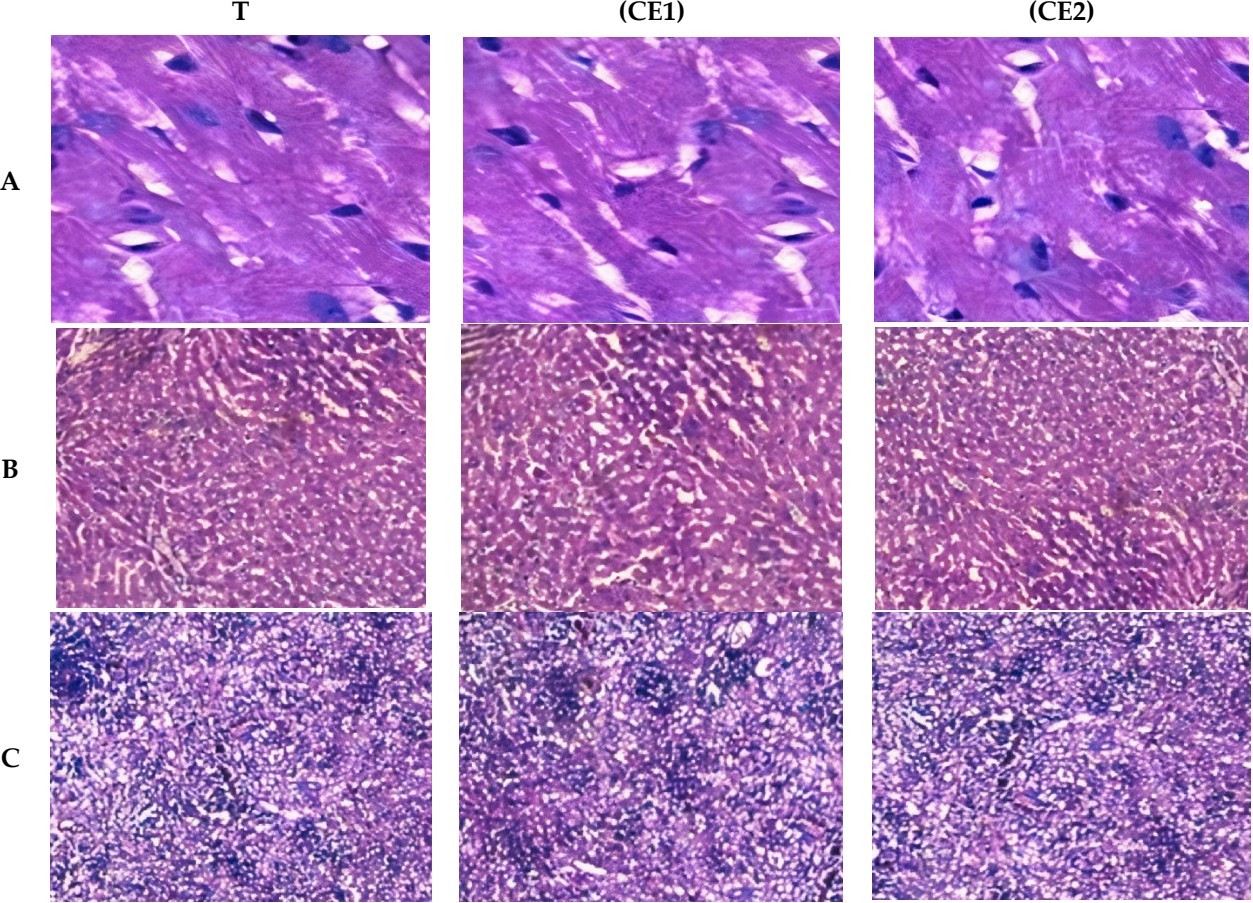

**Figure 1.** *Cont.*

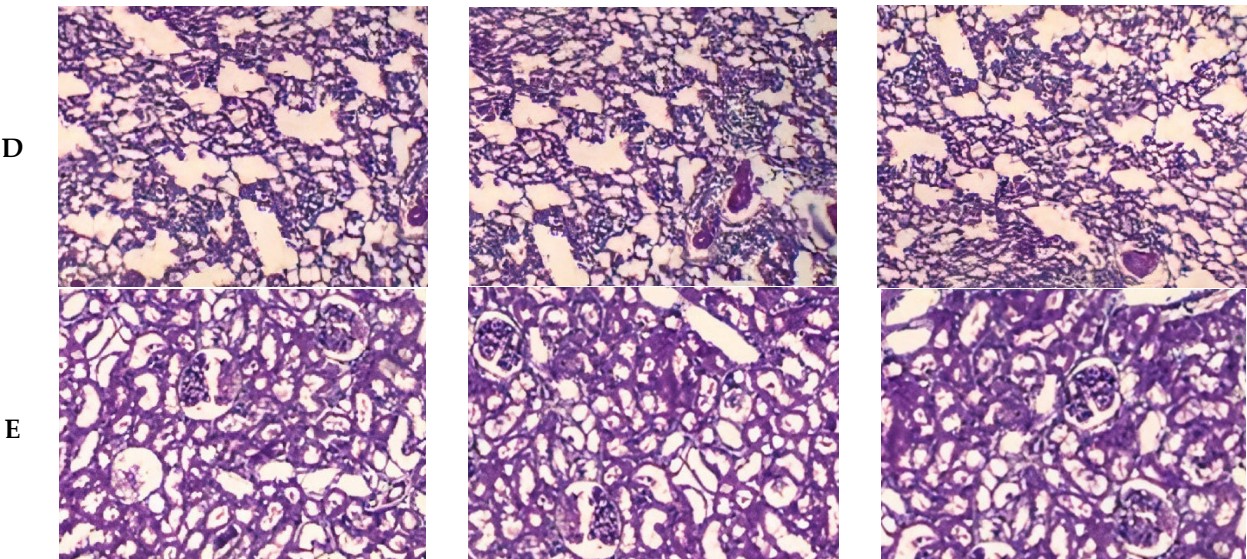

**Figure 1.** Micrography of the heart (**A**), liver (**B**), spleen (**C**), lungs (**D**), and kidneys (**E**) of rats treated with *Catha edulis* Forsk essential oil at doses of 100 and 200 mg/kg (CE1 and CE2) and control rats (T).

### *3.4. In Vivo Activities of Aqueous Extracts and Essential Oils of Catha edulis*

#### 3.4.1. Psychotropic Activity

The effect of the tested essential oil on psychotropic activity was measured by comparing the results obtained in subsequent experiments with those observed for control group and a reference group. Pharmacological tests were carried out by orally administering doses of 100, 200, 400, and 500 mg/kg of *Catha edulis* Forsk essential oil.

#### 3.4.2. Traction Test

The animals that received a dose of 100 mg/kg of *Catha edulis* Forsk essential oil orally quickly recovered without any notable sedative effects. However, at doses of 200 mg/kg, 400 mg/kg, and 500 mg/kg given orally, the mean recovery time decreased significantly compared to the control group (by 0.06 s threshold). These results suggest that *Catha edulis* Forsk essential oil has a remarkable stimulating effect (Table 6).

**Table 6.** The action of *Catha edulis* Forsk essential oil.

| Behavioral Tests | Control | Bromazepam (20 mg/kg; VO) | *Catha edulis* Forsk Essential Oil in (mg/kg; VO) | | | |
|---|---|---|---|---|---|---|
| | | | 100 (mg/kg) | 200 (mg/kg) | 400 (mg/kg) | 500 (mg/kg) |
| Traction | $0.15 \pm 0.05$ s $n = 5$ | $12.00 \pm 0.50$ s $n = 5$ | $0.15 \pm 0.01$ s $n = 5$ | $0.09 \pm 0.02$ s $n = 5$ | $0.09 \pm 0.02$ s $n = 5$ | $0.09 \pm 0.01$ s $n = 5$ |
| Chimney | $5.00 \pm 0.50$ s $n = 5$ | >2 min $n = 5$ | $4.70 \pm 0.20$ s $n = 5$ | $4.30 \pm 0.20$ s $n = 5$ | $3.00 \pm 0.20$ s $n = 5$ | $3.00 \pm 0.20$ s $n = 5$ |
| Hole board | $5.00 \pm 1.00$ $n = 5$ | $0.00 \pm 0.00$ $n = 5$ | $7.00 \pm 1.00$ $n = 5$ | $8.00 \pm 1.00$ $n = 5$ | $10.00 \pm 1.00$ $n = 5$ | $10.00 \pm 1.00$ $n = 5$ |

The results are expressed as mean $\pm$ standard deviation, $p < 0.05$ compared to the control batch.

#### 3.4.3. Chimney Test

The results obtained show that *Catha edulis* Forsk essential oil does not have a sedative effect on mice, but on the contrary, it appears to stimulate their initiative and curiosity. Indeed, mice treated with this essential oil were able to climb the tube, suggesting a stimulating activity of the plant (Table 6).

### 3.4.4. Hole-Board Test

The test shows that oral administration of *Catha edulis* Forsk essential oil at doses of 100, 200, 400, and 500 mg/kg significantly increases the cumulative number of holes explored by mice, reflecting their increased curiosity (Table 6). These results suggest that investigated essential oil stimulates the locomotor activity of mice.

### 3.4.5. Rotarod Test

The results of the rotarod test are presented in Table 7. After oral administration of *Catha edulis* Forsk essential oils for 30, 60, and 120 min, a significant increase ($p < 0.001$) was observed in the average time spent by the animals on the rod as well as their ability to stay on it. These results suggest that, unlike bromazepam, which produces a typical sedative effect, *Catha edulis* Forsk essential oil has a significant stimulating effect.

**Table 7.** The behavior of mice on the rotating rod after oral administration of *Catha edulis* Forsk essential oil and bromazepam (20 mg/kg).

| Behavioral Tests | Dose (mg/kg) | Time in Seconds | | |
| --- | --- | --- | --- | --- |
| | | 30 Min after Administration | 60 Min after Administration | 120 Min after Administration |
| Control | - | $120 \pm 1$ | $120 \pm 1$ | $120 \pm 1$ |
| *Catha edulis* Forsk (VO) | 100 | $125 \pm 2.54$ | $127 \pm 2.46$ | $131 \pm 2.41$ |
| | 200 | $128 \pm 2.84$ | $130 \pm 2.31$ | $135 \pm 2.19$ |
| | 400 | $133 \pm 3.49$ | $136 \pm 3.19$ | $139 \pm 3.15$ |
| | 500 | $140 \pm 3.67$ | $140 \pm 3.94$ | $140 \pm 3.99$ |
| Bromazepam (VO) | 20 | $7 \pm 0.05$ | $22 \pm 0.80$ | $45 \pm 0.3$ |

The results are expressed as mean $\pm$ standard deviation, $p < 0.05$ compared to the control batch.

### 3.5. Antimicrobial Activity

The antimicrobial activity of *Catha edulis* Forsk essential oil was evaluated against several strains of Gram-positive bacteria (*S. pneumoniae* and *S. aureus*) and Gram-negative bacteria (*E. coli, P. mirabilis, P. aeruginosa,* and *E. cloacae*), and the results of the microbial growth inhibition tests are presented in Table 8. The results indicate that the essential oil exhibited effectiveness against all bacterial strains, with a zone of inhibition diameter ranging from 23.78 to 45.33 mm. Dilutions of 1/2, 1/4, 1/8, and 1/16 gave zone of inhibition diameters less than 25 mm, indicating that the concentration of the essential oils is proportional to their activity [51]. The essential oils of *Catha edulis* Forsk demonstrated noteworthy activity against Gram-negative bacteria, with zone of inhibition diameters varying from 1.99 to 31.15 mm. However, it exhibited even greater activity against Gram-positive bacteria, with a zone of inhibition diameter ranging from 2.33 to 45.33 mm. In general, Gram-positive bacteria are more sensitive to the action of essential oils than Gram-negative bacteria.

**Table 8.** Antimicrobial activity of *Catha edulis* Forsk essential oils.

| Strains | Inhibition Zone Diameters in (mm) | | | | |
|---------|------|------|------|------|------|
| | **1/1** | **1/2** | **1/4** | **1/8** | **1/16** |
| *E. coli* | 23.78 ± 1.34 | 11.95 ± 1.03 | 4.88 ± 0.85 | 2.50 ± 0.56 | 1.99 ± 0.21 |
| *P. mirabilis* | 31.15 ± 1.49 | 15.22 ± 1.00 | 10.01 ± 0.65 | 7.5 ± 0.48 | 3.89 ± 0.25 |
| *P. aeruginosa* | 27.65 ± 1.61 | 14.53 ± 0.99 | 8.47 ± 0.75 | 6.1 ± 0.42 | 3.36 ± 0.36 |
| *E. cloaceae* | 31.06 ± 1.34 | 16.41 ± 0.86 | 10.28 ± 0.66 | 7.65 ± 0.38 | 4.99 ± 0.74 |
| *S. pneumoniae* | 25.44 ± 1.86 | 12.84 ± 1.31 | 6.31 ± 0.84 | 3.16 ± 0.57 | 2.33 ± 0.29 |
| *S. aureus* | 45.33 ± 1.96 | 25.25 ± 1.22 | 10.34 ± 0.64 | 9.01 ± 0.37 | 5.24 ± 0.11 |

## 4. Conclusions

This study explored the chemical composition and toxicological properties of the essential oil of *Catha edulis* Forsk from Djibouti. Cathinone and cathine were identified as the main components of the investigated essential oil. Toxicological analysis revealed that the oil has not exhibited toxic effects in animals. Furthermore, it possesses potent antimicrobial properties against both Gram-positive and Gram-negative bacteria, indicating its potential as a supplementary or alternative antimicrobial agent in the treatment of bacterial infections. This work contributes to the current pool of literature data on the pharmacological and toxicological qualities of natural products. It also indicates that the essential oil of *Catha edulis* Forsk from Djibouti has promising antibacterial potential and could be utilized as complementary medicine in the treatment and prevention of antibiotic-resistant bacterial strain infections.

**Author Contributions:** Conceptualization, F.M.A.-L. and T.A.; methodology, A.M.A., and A.A.; software, I.H.A., M.K. and J.M.; validation, F.M.A.-L. and T.A.; formal analysis, M.K., Z.E.M., I.H.A. and A.A; investigation, F.M.A.-L.; resources, F.M.A.-L. data curation, A.M.A. and T.A.; writing—original draft preparation, A.A. and T.A.; writing—review and editing, M.K. and F.M.A.-L.; visualization, A.A.; supervision, F.M.A.-L.; project administration, F.M.A.-L.; funding acquisition, F.M.A.-L. All authors have read and agreed to the published version of the manuscript.

**Funding:** The authors received no external funding for this article.

**Data Availability Statement:** Data is contained within the article.

**Conflicts of Interest:** The authors declare no conflict of interest.

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
