# Peer review of "Chemical Composition of the Essential Oil of Catha edulis Forsk from Djibouti and Its Toxicological Investigations In Vivo and In Vitro"

_processes, doi:10.3390/pr11051324_

Round 1

Reviewer 1 Report

The authors of the present study have designed and conducted the experiments in a good way; however, there are a few major flaws in the presentation of the results, which makes this manuscript unfit for publication.

All the results are given in tables with no indication of significance on them. It is not possible to sort the significant results from the non-significant ones.

In table 7, the time for control is 120± 0 for all 30, 60, and 120 minutes. The reviewer failed to understand the reason why can an animal show such consistent results in each group even without a slight deviation.

The results are rarely supported by reasoning.

A few claims are made in the conclusion part that have no link to the results or any previous section, such as the authors mentioned that " The act of chewing khat may have various negative effects on the body, including hindering the absorption of certain orally administered antibiotics, inducing anorexia and constipation, and even exhibiting toxic effects on the liver. These toxic effects could potentially be attributed to the use of pesticides during the cultivation of khat. Moreover, pregnant women who chew khat are at an increased risk of having low birth weight infants." However, in the results section, the authors mentioned that the effects on the liver were non-significant, and the impact on pregnancy or infant was not observed.

Author Response

Dear Reviewer,

Please find attached the responses to your questions and comments as requested. We appreciate the time and effort you have allocated to reviewing our manuscript, and we hope that our responses have addressed any concerns you may have had. We value your feedback and have made every effort to incorporate your suggestions for scientific level improvement.

Thank you for your continued support and consideration.

Best regards,

All the results are given in tables with no indication of significance on them. It is not possible to sort the significant results from the non-significant ones.

We used a statistical test to determine the uncertainties with a risk alpha of 5%, using the student test and calculating the confidence interval. As our experiments involved comparing weights or performing chronological comparisons, we did not conduct a comparison of means using ANOVA

In table 7, the time for control is 120± 0 for all 30, 60, and 120 minutes. The reviewer failed to understand the reason why can an animal show such consistent results in each group even without a slight deviation.

The Rotarod offers an easy method for testing motor activity in rodents (rats or mice) - an ideal solution for studying central nervous system (CNS) damage, the effects of different pathologies and pharmacological substances on activity motor, resistance to fatigue, etc. Now even easier to use! The new graphical and tactile user interface allows a clear visualization of the speed and times associated with each track. You can now change modes, adjust speed and create protocols directly from the main screen for greater flexibility and maximum functionality.

The animal is placed on the multi-way drum of the ROTAROD and the timer starts. When the animal falls, safely in its own lane, the falling time (in minutes and seconds) and the rotational speed are automatically recorded. A removable upper partition is included to avoid interference between animals running in adjacent lanes.

The ROTAROD is controlled by an advanced microprocessor that offers precise time and speed control. Rotation can be set, electronically, to a constant speed, using a dial on the front panel. The acceleration rate can also be selected at a defined time (30 sec., 1, 2, 5 or 10 min).

A few claims are made in the conclusion part that have no link to the results or any previous section, such as the authors mentioned that " The act of chewing khat may have various negative effects on the body, including hindering the absorption of certain orally administered antibiotics, inducing anorexia and constipation, and even exhibiting toxic effects on the liver. These toxic effects could potentially be attributed to the use of pesticides during the cultivation of khat. Moreover, pregnant women who chew khat are at an increased risk of having low birth weight infants." However, in the results section, the authors mentioned that the effects on the liver were non-significant, and the impact on pregnancy or infant was not observed.

After taking into account your remarks, we revised the conclusion to ensure a strong connection between the results and the final statement.

Reviewer 2 Report

Congratulations !

Author Response

Dear Reviewer,

Thank you for your interest in our manuscript. We greatly appreciate your time and effort in reviewing our work. Please accept our sincere greetings and appreciation for your contribution to the advancement of scientific research.

Best regards, 

Author Response

(The authors gave the same response as above.)

Reviewer 4 Report

Some topics need further clarification:

-change keywords. Try to substitute for terms that are not in the title of the article
-Was the botanical identification of the species carried out? Where can you find the registration number for the specimen?
-for carrying out the tests on animals, was the work submitted to the animal ethics committee?
-How were the slides prepared for the toxicological evaluation of the mice?
-regarding antimicrobial activity, better elucidate the technique for preparing the bacteria and the tested concentrations of the essential oil? Was a stock solution used for the test? clarify better.
-as for the conclusion, it does not bring information related to the results and the objective of the study. The authors work with little elucidated information during the presentation of results and discussion.
-references need proofreading

Author Response

Dear Reviewer,

Thank you for your valuable feedback on our manuscript. We appreciate your time and effort in reviewing our work. Please find below the answers to your questions and comments.

Once again, we thank you for your contribution to the advancement of our scientific research.

Please accept our sincere greetings and gratitude.

Best regards, 

-change keywords. Try to substitute for terms that are not in the title of the article.

We redid the keywords.

-Was the botanical identification of the species carried out? Where can you find the registration number for the specimen?

We have added a paragraph in the part (2.1. Plant Material and Sample Collection) which consists of the identification of the plant species and the registration number at the CERD center.

-for carrying out the tests on animals, was the work submitted to the animal ethics committee?

The Editor communicated to us during the manuscript submission of this issue. hence the Ethical approval for this study was granted by the ethics committee of ESTK-USMS (Morocco); Code 2022/1000.

we added the code in the Section: 2.4.1. Experimental animals

-regarding antimicrobial activity, better elucidate the technique for preparing the bacteria and the tested concentrations of the essential oil? Was a stock solution used for the test? clarify better.

The microbial strains are deposited on a nutrient agar medium by forming streaks. After incubation for 18 hours at 37° C., well-isolated colonies are picked using a platinum loop. These colonies are deposited in 10 ml of sterile physiological water at 0.9%, then the bacterial suspension is homogenized using a vortex. Dilutions are then carried out to standardize the bacterial suspension. The inoculum is adjusted to an optical density of 0.08 to 0.10 at 625 nm, which corresponds to 0.5 McFarland. The final inoculum concentration is 107 CFU/ml.

The aromatogram method described by Jacob and Tonei in 1979 was chosen to evaluate the antibacterial activity of plant extracts. This method consists of using sterile 6 mm paper discs impregnated with different concentrations of extracts. Serial dilutions (1/1, 1/2, 1/4, 1/8 and 1/16) of the essential oil in dimethyl sulfoxide (DMSO) and the aqueous extract in distilled water are made . Sterile Muller-Hinton (MH) agar is poured into sterile 9 cm Petri dishes. 1 ml of inoculum prepared from each strain is evenly spread on the surface of the MH agar. Excess inoculum is removed by aspiration. A volume corresponding to fifty microliters of the dilutions of the extracts is deposited on sterile filter paper discs (Whatman No. 1, 6 mm in diameter). The disks are placed on the surface of the inoculated agar and incubated at 37° C. for 18 hours. After incubation, the effect of the extracts results in the appearance of a transparent circular zone around the disc corresponding to the absence of growth. The larger the diameter of this zone, the more susceptible the strain. Negative controls, DMSO and distilled water, are used to check the growth of the different strains, while ciprofloxacin is used as a positive control. The antibacterial activity is determined using a ruler measuring the diameter of the zone of inhibition (DZI). The results are expressed in DZI and can be symbolized by crosses: not sensitive (-) for D < 8 mm, sensitive (+) for D between 9 and 14 mm, very sensitive (++) for D between 15 and 19 mm and extremely sensitive (+++) for D > 20 mm.

-as for the conclusion, it does not bring information related to the results and the objective of the study. The authors work with little elucidated information during the presentation of results and discussion.

the conclusion has been redone.

-references need proofreading

Ok

Round 2

Reviewer 1 Report

The authors have improved the manuscript; however, the problem of statistical analysis still persists. With thorough comparison (on a statistical basis), it is very difficult to believe the significance of the study. Therefore, it is strongly advised to statistically analyze the results.

Author Response

Dear Reviewer, Following your requested recommendations for more in-depth statistical analyses, we performed these tests in most parts of the study. With a favorable response, please accept our sincere respectful greetings.

Reviewer 4 Report

The authors presented the suggested changes and the article is suitable for publication.

Author Response

Dear Reviewer, We thank you for the scientific interest you have given to our manuscript. With a favorable response, please accept our sincere respectful greetings.

Round 3

Reviewer 1 Report

The authors have significantly improved the manuscript. It could be accepted after making necessary changes regarding language and syntax.